# Increased Risk of Developing Depression in Disability after Stroke: A Korean Nationwide Study

**DOI:** 10.3390/ijerph20010842

**Published:** 2023-01-02

**Authors:** Hea Lim Choi, Kyojin Yang, Kyungdo Han, Bongsung Kim, Won Hyuk Chang, Soonwook Kwon, Wonyoung Jung, Jung Eun Yoo, Hong Jin Jeon, Dong Wook Shin

**Affiliations:** 1Department of Family Medicine/Supportive Care Center, Samsung Medical Center, Sungkyunkwan University School of Medicine, Seoul 06351, Republic of Korea; 2Department of Psychiatry, Depression Center, Samsung Medical Center, Sungkyunkwan University School of Medicine, Seoul 06351, Republic of Korea; 3Department of Statistics and Actuarial Science, Soongsil University, Seoul 06978, Republic of Korea; 4Department of Medical Statistics, The Catholic University of Korea, Seoul 06591, Republic of Korea; 5Department of Physical and Rehabilitation Medicine, Center for Prevention and Rehabilitation, Heart Vascular Stroke Institute, Samsung Medical Center, Sungkyunkwan University School of Medicine, Seoul 06351, Republic of Korea; 6Department of Neurology, Inha University Hospital, Incheon 22332, Republic of Korea; 7Department of Family Medicine, Healthcare System Gangnam Center, Seoul National University Hospital, Seoul 06236, Republic of Korea; 8Department of Health Sciences & Technology, Department of Medical Device Management & Research, and Department of Clinical Research Design & Evaluation, Samsung Advanced Institute for Health Sciences & Technology (SAIHST), Sungkyunkwan University, Seoul 06355, Republic of Korea

**Keywords:** stroke, post-stroke depression, disability, depression

## Abstract

Stroke is a leading cause of mortality and a major cause of disability worldwide. A significant number of stroke survivors suffer from depression, impeding the activities of daily living and rehabilitation. Here, we examined the risk of depression among stroke survivors according to the severity of disabilities and compared its incidence with a matched control group. We included data from the Korean National Health Insurance Service of 207,678 stroke survivors. Cox proportional hazard models were used to calculate the risk of depression among stroke survivors. Stroke survivors had a greater risk of developing depression than the matched control group with an adjusted hazard ratio of 2.12 (95% confidence interval 2.09–2.15). Stroke survivors with more severe disabilities were associated with a higher risk of depression than those with mild disabilities. The risk of developing depression was prominently high within the first year after a stroke. Males and younger people (<65 years) were independent risk factors for depression in stroke survivors. This study demonstrated an increased risk of developing depression in stroke survivors compared to control subjects, and a higher risk of depression was associated with a more severe degree of disability. Clinicians should be aware of the risk of depression developing in stroke survivors, especially those with disabilities.

## 1. Introduction

Stroke is a leading cause of death worldwide [1] and the second leading cause of disability-adjusted life years in Korea [2]. A significant proportion of stroke survivors have disabilities after a stroke. A meta-analysis conducted in the United States found that the prevalence of disability after a stroke was between 24% and 49% [3]. As survival after a stroke increases due to early intervention and advanced post-stroke management, the management of additional complications after stroke is becoming an important issue. Stroke survivors experience not only physical handicaps but also neuropsychiatric disorders, such as cognitive disorder, anxiety disorder, and depression [4,5]. Post-stroke depression (PSD) is among the most common neuropsychiatric consequences, and approximately 30% of stroke survivors experience depression [4]. PSD increases functional and cognitive impairment and impedes daily living activities and rehabilitation [4,5]. A recent meta-analysis, which included 30 hospital-based studies, 19 rehabilitation-based studies, and 13 population-based studies [6], showed that depression was present in 31% of stroke survivors at any time up to 5 years after a stroke [6]. A large population-based cohort study conducted in Denmark reported a 24.5% incidence of depression in 34,346 registered stroke patients at any time within 2 years, which was an 8.99 times higher risk of developing depression than the reference group [7].

Most of the previous studies are from Western countries and are either hospital-based studies [8,9,10] or longitudinal studies with short-term follow-up periods [11,12,13,14]. Moreover, even though stroke survivors often experience disability at different degrees of severity after a stroke event, no studies have considered the impact of this on the prevalence of depression among stroke survivors.

In this retrospective cohort study that uses a nationwide health claims database with long-term follow-up data, we examined the risk of depression among stroke survivors according to the severity of disability and observed its incidence compared with the matched control group.

## 2. Materials and Methods

### 2.1. Data Source

This study was based on data provided by the Korean National Health Insurance Service (KNHIS). The KNHIS is a single provider of health care to 97% of the Korean population, offering medical aid to the remaining 3% of the population with the lowest income [15]. The KNHIS provides a biennial health screening program to all Koreans aged older than 40 years and to all employees regardless of age [16]. From this program, the KNHIS collects extensive information on anthropometric measurements (height, weight, blood pressure, etc.), health behaviors (smoking status, alcohol consumption, etc.), and laboratory results (lipid profiles, blood glucose, etc.) [17,18]. Along with data from the health screening program, KNHIS also gathers the medical information of each individual administered in this database on the use of medical facilities and prescriptions with the International Classification of Disease, 10th revision (ICD-10) diagnosis codes.

### 2.2. Study Population

This retrospective cohort study included a total of 800,646 patients who experienced a stroke during the period from 1 January 2010 to 31 December 2018. Stroke was defined according to the recording of ICD-10 codes I60−I64 during hospitalization with claims for brain resonance imagining or brain CT scan [19,20,21]. Participants younger than 20 years (n = 8) and those who did not undergo a health screening exam within 2 years (n = 465,257) were excluded from this study. Those diagnosed with a depressive disorder (F32–34, F39) before stroke diagnosis (n = 116,375) and those with missing information (n = 11,328) were also excluded. This resulted in a total of 207,678 stroke patients who were randomly matched 1:1 to a control population (n = 294,506) with respect to age, sex, and index year without replacement. The description of selecting study population is illustrated in Figure 1. Matching was serially performed year by year to ascertain that stroke survivors diagnosed during a specific year match control subjects alive in the same year based on age and sex. The index date of control subjects corresponded to the date of stroke diagnosis of matched stroke survivors. We applied the same exclusion criteria in the selection of control subjects.

The study was approved by the institutional review board of Samsung Medical Center (IRB No.2020-12-068). The requirement for informed consent of participants was waived as the study used de-identified data from an administrative database.

### 2.3. Assessment of Stroke Severity

The severity of stroke was determined by disability registration, which was established in 1989 by the Korean Government to provide welfare benefits according to the type and severity of disability [22,23]. The registration process requires the results of a disability diagnosis determined by a physician in the specialized field of concern according to standardized criteria. The severity of disability was also determined by a physician in a specialized field based on the guidelines provided by the National Health Service. The degree of disability was described by levels, from 1 (the most severe) to 6 (the least severe). Each type of disability has specific criteria to determine the degrees of severity.

In this study, we used the criteria for ‘disability due to brain injury’, and its severity was assessed by rehabilitation medicine physicians, neurologists, or neurosurgeons. The degree of disability severity due to brain injury was graded from 1 to 6, which was determined by the Barthel index score: Grade 1: ≤32, Grade 2: 33–54, Grade 3: 54–79, Grade 4: 70–80, Grade 5: 81–90, Grade 6: ≥91. We defined Grades 1 to 3 as severe disability and Grades 4 to 6 as mild disability (Appendix A).

### 2.4. Study Outcomes and Follow-Up

The endpoint of the study was newly diagnosed depression (F32–34, F39), as used in previous studies [24,25,26]. The study subjects were followed from the date of the stroke diagnosis to the date of depression diagnosis, death or outmigration, or the end of the follow-up period (until December 31, 2019), whichever came first.

### 2.5. Covariates

The place of residence was categorized into metropolitan, urban, and rural areas, and the income level was categorized into quartiles, in which the medical-aided subjects were merged into the lowest-income quartile group. Information on health behavior was obtained from standardized self-questionnaires. Smoking status was classified as current, past, and never smoker, and alcohol consumption was classified as heavy (>30 g/day), mild (0–30 g/day), and non-drinker. Regular exercise was defined as exercising with moderate intensity for more than 30 min at least 5 times a week or with a high intensity for more than 20 min at least 3 times a week. Information on the history of comorbidities, such as diabetes mellitus, hypertension, and dyslipidemia, was based on ICD-10 codes, medical prescriptions, and health screening results. The Charlson Comorbidity Index (CCI) was calculated based on ICD-10 codes to assess the overall comorbidity load [27].

### 2.6. Statistical Analysis

We used Student’s t-tests for continuous variables and χ^2^ test for categorical variables to describe baseline characteristics of study participants. Continuous variables are presented as mean ± standard deviation and categorical variables as numbers (percentages). The stroke survivor group and matched control group were compared, and the stroke survivor group was further sub-grouped into with or without disabilities and compared with a matched control group.

The incidence rate (IR) of depression was calculated as the number of diagnosed depression events per 1000 person-years with 95% confidence intervals. The risk of depression events in the control group and stroke survivors with or without disabilities was calculated using a Cox proportional hazards regression analysis. In Model 1, covariates such as age, sex, and CCI were adjusted, and Model 2 included covariates adjusted in Model 1. Moreover, place of residence, income quartile, diabetes mellitus (DM), hypertension (HTN), dyslipidemia, smoking status, alcohol consumption, and regular exercise were further adjusted. We further examined the different risks for depression among stroke survivors with different degrees of disability severity compared to the matched control group. To observe the differences in the risk for developing depression based on stroke survival period, we repeated analyses with a 1-year follow-up period, after a 1-year lag period, and after a 5-year lag period, accordingly. All statistical analyses were performed by SAS version 9.4 (SAS institute, Cary, NC, USA), with *p*-values < 0.05 considered statistically significant.

## 3. Results

### 3.1. Baseline Demographics

The baseline characteristics of the study population are summarized in Table 1. A total of 207,678 stroke survivors (without disability n = 185,183, with disability n = 22,495) and 294,506 matched control subjects were analyzed. Stroke survivors had a higher prevalence of hypertension (76.1% vs. 49.3%, *p* < 0.001), DM (29.4% vs. 17.7%, *p* < 0.001), and dyslipidemia (66.8 vs. 33.2%, *p* < 0.001) than matched stroke subjects. The proportion of current smokers was lower in the stroke survivor group than in the control group (54.6% vs. 58.6%, *p* < 0.001). Stroke survivors had higher BMI, systolic blood pressure, diastolic blood pressure, serum glucose level, total cholesterol, and triglyceride compared to the control group (all *p* < 0.001). Stroke survivors with disability tended to be older than those without disability or the control group (66.42 ± 10.61 vs. 64.38 ± 12.26 vs. 64.56 ± 12.2 years, respectively, *p* < 0.001).

### 3.2. Risk of Depression in Stroke Survivors Compared to Matched Control Subjects

Table 2 presents the risk for depression presented as HR and 95% CI. Stroke survivors had a greater risk for depression than matched control subjects, even after adjustment for covariates (adjusted HR [aHR] 2.12, 95% CI 2.09–2.15). Among stroke survivors, those with severe disability were associated with a higher risk of depression (aHR 3.39, 95% CI 3.28–3.51) than those with a mild disability (aHR 2.46, 95% CI 2.39–2.53) compared to the matched control group.

### 3.3. Risk of Depression in Stroked Survivors by Survival Periods

Stroke survivors had a 5-times-higher risk of developing depression during the first year after stroke diagnosis compared to matched control subjects (aHR 5.02, 95% CI 4.89–5.16). As the time lag period was followed, the relative risk of developing depression decreased with an aHR of 1.37 (95% CI 1.34–1.40) after a 1-year lag but remained slightly higher than the matched control group even after a 5-year lag (aHR 1.17, 95% CI 1.12–1.21). Stroke survivors with a disability exhibited a higher risk of depression than those without disabilities and a similar pattern of change in aHR was found throughout the lag period. Comparing stroke survivors according to the degree of disability severity, the risk for developing depression increased as severity increased compared to the matched control group one year after stroke (severe disability aHR 9.29, 95% CI 8.81–9.73; mild disability aHR 6.09, 95% CI 5.84–6.35; no disability aHR 4.79, 95% CI 4.67–4.92) (Table 3). The incidence probability of depression in stroke survivors and control subjects is shown in Figure 2.

### 3.4. Risk of Depression in Stroked Survivors by Age and Sex

After stratifying analyses according to age and sex, the association between stroke and the risk of developing depression was more evident in younger (aged < 65; aHR 5.39, 95%CI 5.12–5.67, *p* for interaction < 0.001) and male (aHR 3.78, 95% CI 3.62–3.95, *p* for interaction < 0.001) participants. In addition, there were also increasing trends in the risk of depression according to the severity of disability among stroke survivors, regardless of age and sex (Table 4).

## 4. Discussion

To the best of our knowledge, this is the first nationwide large population-based cohort study in Korea to examine the risk of depression among stroke survivors according to the severity of disability and length of survival with up to a 5-year lag period. A major strength of this study is that a large sample was collected from national claim data with representativeness. Moreover, we compared the risk of developing depression according to the degree of disability severity after stroke, which was evaluated with a standardized criterion applied equally to stroke survivors.

It is known that the delayed rehabilitation of stroke survivors is a risk factor for PSD [28,29,30]. Moreover, depression itself can impede the recovery of stroke survivors [28,31]. A statement from the American Heart Association/American Stroke Association [32] and guidelines for adult stroke rehabilitation and recovery [33] recommend that stroke patients are screened for PSD before discharge using tools such as the Patient Health Questionnaire 2. Moreover, previous studies have shown that the early remission of PSD after stroke was associated with greater recovery in activities of daily living (ADL) function, suggesting that early and effective treatment of depression is recommended to stroke survivors [34,35]. Some researchers tried to observe the effect of interventional treatment on the frequency of depressive symptoms in stroke patients, but treatments such as thrombolytic therapy in ischemic stroke [36] or hematoma evacuation in hemorrhagic stroke [37] showed no clear association with the prevalence of PSD. On the other hand, encouraging stroke survivors to engage in early rehabilitation activities was associated with a decreased prevalence of PSD [38]. Therefore, to prevent stroke survivors from retaining ADL by being trapped in a vicious cycle, early rehabilitation should be provided to stroke survivors within an early period of recovery time. Since stroke survivors with depression are susceptible to experiences of hopelessness, guilt, or even suicidal ideation, a referral to a psychiatric specialist should not be overlooked [28].

We found that stroke survivors had a greater risk of depression than matched control subjects. This result was comparable to other population-based studies where the risk of depression among stroke patients was about two- to four-fold higher than the reference group [7,39,40]. The etiology of PSD is reported to be heterogeneous and can be explained by the following pathophysiological mechanisms [41,42]: (1) Ischemic injury to the brain can lead to reduced levels of monoamines, such as serotonin, noradrenaline, and dopamine, leading to depressed mood and impaired cognition [5,41,43]. (2) The abnormal response of neurotrophic factors, such as the brain-derived neurotrophic factor, which is known to protect neurons after ischemic injury of the brain, is thought to increase the risk of PSD [44,45]. (3) The chronic inflammation and dysregulation of the hypothalamic–pituitary–adrenal (HPA) axis after stroke is suspected to increase the risk of PSD by intimidating neurogenesis, particularly in the hippocampus and frontal cortex [5,46,47,48]. (4) An increased level of glutamate, leading to excitotoxicity is associated with the development of PSD [49,50]. Overall, PSD is caused by a complex and interconnected mechanism that still needs further investigation.

To the best of our knowledge, the current study is the first investigation to show that stroke survivors with a more severe degree of disability are at a higher risk of PSD, highlighting the importance of elucidating underlying mechanisms through further research. Moreover, the prevalence of depression was higher in post-stroke patients compared with orthopedic patients with similar physical impairments [51]. This suggests that the biological mechanism disturbed by stroke contributes to the development of depression. Most of the proposed mechanisms attribute PSD to the aforementioned reduction in monoamines and excitotoxicity mediated by increased glutamate levels [52]. A higher infarct volume results in a higher glutamate level in stroke patients [53]. It could be hypothesized that severely disabled post-stroke survivors are more vulnerable to reduced serotonin levels and enhanced glutamate-mediated excitotoxicity, leading to a greater change in mood and neuronal death. To further elucidate the correlation between brain damage after stroke and depressive symptoms, investigators explored the relationship between changes in gray matter volumes and depression in post-stroke patients and found a significant decrease in the gray matter volume of the left middle frontal gyrus (MFG) [54,55,56]. The MFG is known as a mediator of emotion and cognitive functions; therefore, the decrease in its volume is expected to disturb emotional control and result in a depressive mood [57,58,59,60].

In this study, HR was highest in the first year after stroke and declined after 1-year and 5-year lags. This is consistent with the results of a meta-analysis, which revealed the frequency of depression was highest in the first year (29%) and declined afterward [61]. A longitudinal study of up to 15 years to observe the natural history of depression, reported that the prevalence of depression ranged from 29% to 39% and mostly started within a year of the stroke, with one-third of the cases starting in the first 3 months after stroke [62]. A similar result was reported in a Danish study, where the risk of depression was highest after 3 months (HR 8.99) and declined as the follow-up period reached 2 years (HR 1.93) [7]. Despite the decline in the incidence of PSD as time passes, our results indicate that the risk for newly diagnosed PSD exists, even after the first year, is about 5% per year. Therefore, it is important to look after stroke survivors due to their depressive symptoms within the first year after the stroke and continue surveillance for PSD afterward.

In the current study, male stroke patients were at higher risk of developing depression compared to matched control subjects than female stroke patients. However, most of these studies either did not find a correlation between gender and PSD [5,28] or suggested that females were more prone to developing PSD due to hormonal changes [63,64]. Our results could be somewhat influenced by the expected role of men in Asian cultures, where masculine identity is prominent. One study reported that a high level of masculine centrality was strongly associated with psychological distress [65]. Moreover, in Korea, the heads of the household are predominantly male, and anything that can intimidate their job security could provoke psychological stress. A study on Korean employees reported that the major causes of stress among male employees are the job itself and facing financial problems [66]. Therefore, we could infer that male stroke survivors are at a higher risk of developing depression because post-stroke disabilities could threaten their masculine centrality and financial status.

Our study showed a higher risk of depression in younger stroke patients than in older stroke patients. The association between age and PSD is controversial [5,28]. Some studies reported that stroke patients aged under 70 years had a greater risk of depression since it could be more distressful for younger patients to confront physical disability after a stroke [28,63,67]. However, other studies reported that depression was more prevalent among older stroke survivors [28,68]. In line with males being the predominant portion of the employee population, the majority of stroke patients under the age of 65 years in our study were in the working population; thus, post-stroke sequelae in young stroke survivors could intimidate job security resulting in a higher risk for developing depression than those older than 65 years.

Overall, our findings and those from other studies suggest that more meticulous management of depression should be given to stroke survivors with physical conditions that require the help of others, especially during the early period. Our study also implies that the risk of developing depression in those who are male and younger should be given greater consideration in the guidelines for managing PSD and requires further research to establish a consensus [31].

This study has several limitations. Firstly, since we used administrative data, detailed clinical information regarding the depression state, type, and location of the stroke was not provided. Secondly, there could be a detection bias where stroke survivors tend to visit hospitals more often than control subjects, and thus depression has a greater likelihood of being detected. Lastly, our results are limited in terms of ethnicity since all the participants included in the KNHIS are Korean.

## 5. Conclusions

In conclusion, the current study demonstrated an increased incidence of developing depression in stroke survivors compared to control subjects. Stroke survivors with severe disabilities had a greater risk of depression compared to those with mild or no disabilities. A psychological assessment with appropriate counseling is needed for the potential presence of depression in stroke survivors to prevent the development of PSD and provide early intervention with rehabilitation.

## Figures and Tables

**Figure 1 ijerph-20-00842-f001:**
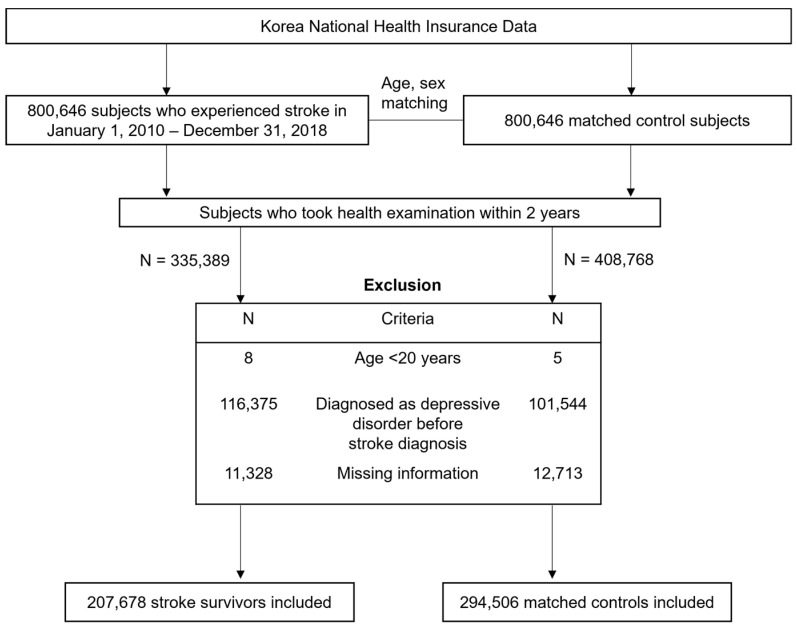
Flowchart of selecting study participants.

**Figure 2 ijerph-20-00842-f002:**
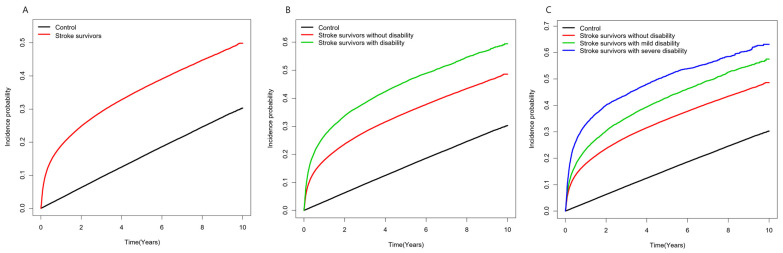
Incidence probability of the depression in (**A**) stroke survivors and control group (**B**) stroke survivors with disability, without disability, and control group; (**C**) stroke survivors without disability, with disability by the severity degrees, and control group.

**Table 1 ijerph-20-00842-t001:** Baseline characteristics of study participants.

	Control Group	Stroke Survivors	*p*-Value	Control Group	Stroke Survivors without Disability	Stroke Survivors with Disability	*p*-Value
	(N = 294,506)	(N = 207,678)		(N = 294,506)	(N = 185,183)	(N = 22,495)	
Age	64.56 ± 12.2	64.6 ± 12.11	0.268	64.56 ± 12.2	64.38 ± 12.26	66.42 ± 10.61	<0.001
Sex, male	182,928 (62.1)	129,674 (62.4)	0.019	182,928 (62.1)	115,021 (62.1)	14,653 (65.1)	<0.001
Income, low 25%	55,466 (18.8)	44,770 (21.6)	<0.001	55,466 (18.8)	39,216 (21.2)	5554 (24.7)	<0.001
Place, urban	129,826 (44.1)	84,428 (40.7)	<0.001	129,826 (44.1)	75,758 (40.9)	8670 (38.5)	<0.001
Regular exercise	65,370 (22.2)	38,795 (18.7)	<0.001	65,370 (22.2)	34,908 (18.9)	3887 (17.3)	<0.001
Follow up duration	4.89 ± 2.59	3.31 ± 2.8	<0.001	4.89 ± 2.59	3.33 ± 2.81	3.19 ± 2.77	<0.001
DM	52,235 (17.7)	60,984 (29.4)	<0.001	52,235 (17.7)	53,423 (28.9)	7561 (33.6)	<0.001
HTN	145,176 (49.3)	158,136 (76.1)	<0.001	145,176 (49.3)	139,678 (75.4)	18,458 (82.1)	<0.001
Dyslipidemia	97,759 (33.2)	138,773 (66.8)	<0.001	97,759 (33.2)	123,697 (66.8)	15,076 (67.0)	<0.001
Smoking			<0.001				<0.001
None	172,707 (58.6)	113,310 (54.6)		172,707 (58.6)	100,907 (54.5)	12,403 (55.1)	
Ex-	66,065 (22.4)	37,369 (18.0)		66,065 (22.4)	33,245 (18.0)	4124 (18.3)	
Current	55,734 (18.9)	56,999 (27.5)		55,734 (18.9)	51,031 (27.6)	5968 (26.5)	
Alcohol consumption			<0.001				<0.001
None	174,887 (59.4)	122,215 (58.9)		174,887 (59.4)	108,200 (58.4)	14,015 (62.3)	
Mild	98,326 (33.4)	66,813 (32.2)		98,326 (33.4)	60,287 (32.6)	6526 (29.0)	
Heavy	21,293 (7.2)	18,650 (9.0)		21,293 (7.2)	16,696 (9.0)	1954 (8.7)	
CCI	1.58 ± 1.75	4.15 ± 2.31	<0.001	1.58 ± 1.75	4.15 ± 2.31	4.57 ± 2.37	<0.001
BMI	24.07 ± 3.07	24.22 ± 3.24	<0.001	24.07 ± 3.07	24.22 ± 3.24	24.44 ± 3.33	<0.001
Waist circumference	83.07 ± 8.53	83.83 ± 8.7	<0.001	83.07 ± 8.53	83.83 ± 8.7	85.03 ± 8.72	<0.001
Glucose	103.47 ± 26.1	110.35 ± 37.96	<0.001	103.47 ± 26.1	110.35 ± 37.96	111.34 ± 38.29	<0.001
SBP	127.12 ± 15.33	132.28 ± 17.74	<0.001	127.12 ± 15.33	132.28 ± 17.74	132.99 ± 17.68	<0.001
DBP	77.45 ± 9.84	80.54 ± 11.52	<0.001	77.45 ± 9.84	80.54 ± 11.52	80.64 ± 11.35	<0.001
Total cholesterol	195 ± 38.12	197.89 ± 40.78	<0.001	195 ± 38.12	197.89 ± 40.78	196.38 ± 40.6	<0.001
TG	117.8 (117.58–118.03)	126.64 (126.35–126.93)	<0.001	117.8 (117.58–118.03)	126.43 (126.12–126.74)	128.42 (127.54–129.3)	<0.001
GFR	85.68 ± 43.19	83.92 ± 40.03	<0.001	85.68 ± 43.19	83.92 ± 40.03	84.02 ± 42.15	<0.001

Abbreviations: DM, diabetes mellitus; HTN, hypertension; CCI, Charlson Comorbidity Index; BMI, body mass index; SBP, systolic blood pressure; DBP, diastolic blood pressure; TG, triglyceride; GFR, glomerular filtration rate.

**Table 2 ijerph-20-00842-t002:** Comparison of the risk of developing depression in stroke survivors and matched control group.

	N	IR (per 1000 PYs)	HR (95% CI)		
			Unadjusted	Adjusted (Model 1)	Adjusted (Model 2)
Comparison between stroke survivors and control group					
Control	294,506	34.40	1 (ref.)	1 (ref.)	1 (ref.)
Stroke survivors	207,678	98.05	2.69 (2.66–2.73)	2.21 (2.18–2.24)	2.12 (2.09–2.15)
Comparison between stroke survivors with, and without disability, and control group					
Control	294,506	34.40	1 (ref.)	1 (ref.)	1 (ref.)
No disability after stroke	185,183	93.28	2.57 (2.54–2.60)	2.13 (2.10–2.16)	2.04 (2.02–2.07)
Disability after stroke	22,495	138.95	3.77 (3.69–3.85)	2.87 (2.80–2.93)	2.74 (2.68–2.81)
Comparison between stroke survivors without disability, with disability by the severity degrees, and control group					
Control	294,506	34.40	1 (ref.)	1 (ref.)	1 (ref.)
No disability after stroke	185,183	93.29	2.57 (2.54–2.60)	2.13 (2.11–2.16)	2.05 (2.02–2.07)
Mild disability after stroke	14,705	123.69	3.39 (3.30–3.48)	2.57 (2.50–2.65)	2.46 (2.39–2.53)
Severe disability after stroke	7790	173.91	4.60 (4.45–4.75)	3.53 (3.41–3.65)	3.39 (3.28–3.51)

Model 1: Mutually adjusted for age, sex, CCI; Model 2: Age, sex, CCI, place, income, DM, HTN, dyslipidemia, smoking, alcohol consumption, exercise. Abbreviations: IR, incidence rate; HR, hazard ratio; CI, confidence interval; CCI, Charlson Comorbidity Index; DM, diabetes mellitus; HTN, hypertension; PYs, person-years.

**Table 3 ijerph-20-00842-t003:** Comparison of the risk of developing depression in stroke survivors and matched control group.

	First 1 Year		After 1-Year Lag		After 5-Year Lag	
	IR (per 1000 PYs)	aHR (95% CI)	IR (per 1000 PYs)	aHR (95% CI)	IR (per 1000 PYs)	aHR (95% CI)
Comparison between stroke survivors and control group						
Control	31.69	1 (ref.)	35.08	1 (ref.)	37.59	1 (ref.)
Stroke survivors	220.73	5.02 (4.89–5.16)	58.93	1.37 (1.34–1.39)	48.64	1.17 (1.12–1.21)
Comparison between stroke survivors with, and without disability, and control group						
Control	31.69	1 (ref.)	35.08	1 (ref.)	37.59	1 (ref.)
No disability after stroke	207.61	4.79 (4.67–4.92)	57.07	1.34 (1.32–1.37)	47.61	1.16 (1.11–1.20)
Disability after stroke	328.26	7.14 (6.89–7.40)	75.15	1.58 (1.53–1.63)	57.81	1.25 (1.15–1.35)
Comparison between stroke survivors without disability, with disability by the severity degrees, and control group						
Control	31.69	1 (ref.)	35.08	1 (ref.)	37.59	1 (ref.)
No disability after stroke	207.61	4.79 (4.67–4.92)	57.07	1.34 (1.32–1.37)	47.61	1.16 (1.11–1.20)
Mild disability after stroke	278.46	6.09 (5.84–6.35)	73.58	1.54 (1.48–1.60)	59.38	1.28 (1.17–1.40)
Severe disability after stroke	431.83	9.29 (8.87–9.73)	78.87	1.67 (1.58–1.77)	53.79	1.17 (1.01–1.36)

aHRs were adjusted for age, sex, CCI, place, income, DM, HTN, dyslipidemia, Smoking, alcohol consumption, exercise. Abbreviations: IR, incidence rate; PYs, person-years; aHR, adjusted hazard ratio; CI, confidence interval; CCI, Charlson Comorbidity Index.

**Table 4 ijerph-20-00842-t004:** Risk of depression in stroke survivors and control group according to age and sex.

		**N**	**Depression Event**	**IR** **(per 1000 PYs)**	**aHR** **(95% CI)**	***p* for** **Interaction**
By age						
Age < 65	Control	137,526	14,432	20.93	1 (ref.)	<0.001
	No disability after stroke	87,533	24,556	76.36	2.72(2.66–2.78)	
	Mild disability after stroke	5511	2237	109.39	3.66 (3.50–3.83)	
	Severe disability after stroke	3323	1669	170.62	5.39 (5.12–5.67)	
Age ≥ 65	Control	156,980	35,148	46.76	1 (ref.)	
	No disability after stroke	97,650	32,944	111.75	1.76 (1.73–1.79)	
	Mild disability after stroke	9194	3940	133.6	2.04 (1.98–2.12)	
	Severe disability after stroke	4467	2124	176.58	2.62 (2.51–2.74)	
		**N**	**Depression Event**	**IR** **(per 1000 PYs)**	**aHR** **(95% CI)**	**p for** **Interaction**
By sex						
Male	Control	182,928	25,659	28.49	1 (ref.)	<0.001
	No disability after stroke	115,021	32,899	83.95	2.19 (2.15–2.23)	
	Mild disability after stroke	9532	3734	113.65	2.73 (2.63–2.83)	
	Severe disability after stroke	5121	2401	164.12	3.78 (3.62–3.95)	
Female	Control	111,578	23,921	44.24	1 (ref.)	
	No disability after stroke	70,162	24,601	109.58	1.89 (1.86–1.93)	
	Mild disability after stroke	5173	3734	142.99	2.16 (2.07–2.25)	
	Severe disability after stroke	2669	2401	193.84	2.92 (2.76–3.08)	

aHRs were adjusted for age, sex, CCI, place, income, DM, HTN, dyslipidemia, smoking, alcohol consumption, and exercise. Abbreviations: IR, incidence rate; PYs, person-years; aHR, adjusted hazard ratio; CI, confidence interval; CCI, Charlson Comorbidity Index.

## Data Availability

The data of this study are available from KNHIS. Restrictions apply to the availability of these data, which were used under license for this study. Data are available for the authors with the permission of the KNHIS.

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
