# Peer review of "Increased Risk of Developing Depression in Disability after Stroke: A Korean Nationwide Study"

_ijerph, 2023, doi:10.3390/ijerph20010842_

Round 1
Reviewer 1 Report
Thank you for well prepared manuscript. You have evaluated the depression in stroke patients that is an important and often overlooked issue. The manuscript it scientifically sound with appropriate methodology. Additionally, issues are well explained in the limitations section. There are only few issues I believe could be resolved.
1. As you were using some parametric statistical methods, I believe the distributions were normal even though this could be stated somewhere.
2. I miss some more information about the 1:1 matching. What algorithms were used to randomly perform the matching?
3. When comparing groups some post-hoc corrections like Bonferroni could be used even though in this number I believe the outcomes would not be changed.
Reviewer 2 Report
An interesting work on the "dark side of the moon" - that is, on incomplete recovery after a stroke among patients treated with interventions or not. As in cardiology, the extensive use of interventional treatment increased survival among patients with myocardial infarction, but at the same time it significantly increased the number of patients with large, post-infarction left ventricular damage, who previously died due to rapidly progressing heart failure or arrhythmias, and are now the most frequently hospitalized patients, requiring complex treatment for heart failure or dangerous arrhythmias and implantation of devices to prevent sudden cardiac death.
In the study, the authors, based on the results of over 200,000 patients, found that stroke in young patients (<65 years of age), with significant neurological damage, is associated with a high risk of developing depression. And just like in cardiology, we hope that in the coming years supportive treatment will be started, not only pharmacological, but also aimed at restoring the "movement" abilities of these patients.
It is an interesting study of the frequency of stroke sequelae, both in the interventional and conservative treatment group. A very large population covered by the study increases the statistically significant truth of the conclusions. In the studies on the fate of patients after surgery, the analysis mainly covers neurological symptoms, including patients' independence (NIHSS and mRSA scale), less frequently, psychological and psychiatric complications associated with a very serious vascular incident are assessed.
I would be very curious whether interventional treatment reduces the frequency of e.g. depression, or maybe on the contrary - partial improvement after surgery leads to a state in which the patient assesses his health better than the patient without interventional treatment?
